# FASTVMT⚡: ELIMINATING REDUNDANCY IN VIDEO MOTION TRANSFER

**Yue Ma [2], Zhikai Wang [1†], Tianhao Ren [1†], Mingzhe Zheng [2], Hongyu Liu [2], Jiayi Guo [3], Kunyu Feng, Yuxuan Xue [4], Zixiang Zhao [5], Konrad Schindler [5], Qifeng Chen [2], Linfeng Zhang [1✉]**

[1] EPIC Lab, Shanghai Jiao Tong University, [2] HKUST, [3] THU, [4] Meta, [5] ETH Zürich

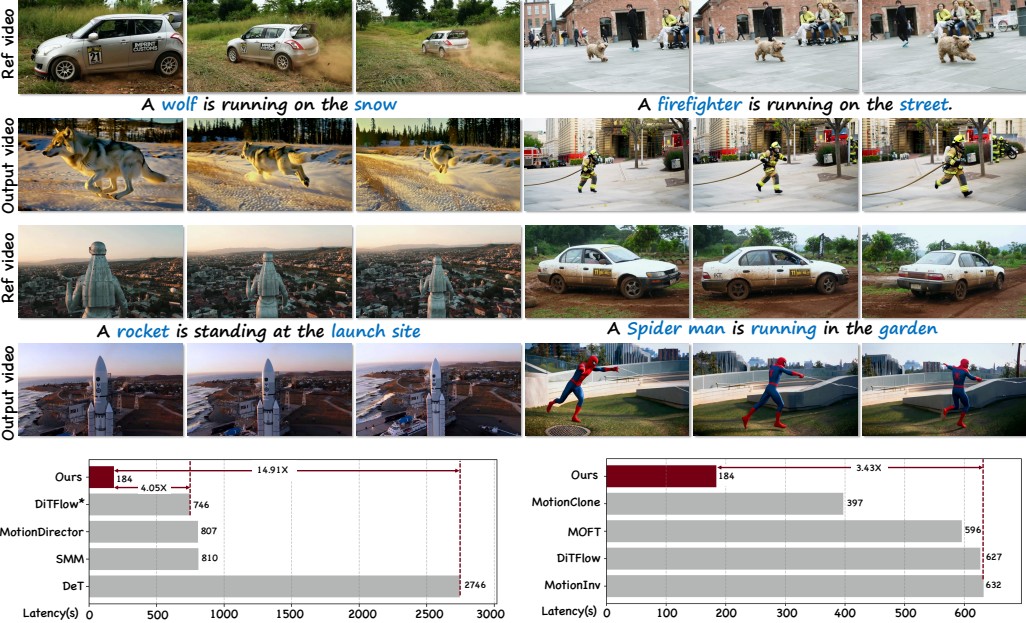

Figure 1: **Efficient motion transfer with FastVMT:** By eliminating redundant attention computations and reusing previously computed gradients, we achieve faster motion transfer for single-as well as multi-object motion, camera ego-motion, and complex articulations.

## ABSTRACT

Video motion transfer aims to synthesize videos by generating visual content according to a text prompt while transferring the motion pattern observed in a reference video. Recent methods predominantly use the Diffusion Transformer (DiT) architecture. To achieve satisfactory runtime, several methods attempt to accelerate the computations in the DiT, but fail to address structural sources of inefficiency. In this work, we identify and remove two types of computational redundancy in earlier work: ***motion redundancy*** arises because the generic DiT architecture does not reflect the fact that frame-to-frame motion is small and smooth; ***gradient redundancy*** occurs if one ignores that gradients change slowly along the diffusion trajectory. To mitigate motion redundancy, we mask the corresponding attention layers to a local neighborhood such that interaction weights are not computed unnecessarily distant image regions. To exploit gradient redundancy, we design an optimization scheme that reuses gradients from previous diffusion steps and skips unwarranted gradient computations. On average, FastVMT achieves a $3.43\times$ speedup without degrading the visual fidelity or the temporal consistency of the generated videos.

---

† Equal contribution
✉ Corresponding author

# 1 INTRODUCTION

Motion transfer aims to generate a novel video by transferring the dynamics of a reference video sequence to a target sequence, while preserving the target's appearance and semantics. For instance, the reference video might show an action sequence performed by an actor, which shall be transferred to a target subject while preserving their identity; or the reference might prescribe a particular camera path through the scene, which one would like to replicate for the target scene (see Fig. 1). In other words, motion transfer offers an intuitive interface for controllable motion synthesis, with applications ranging from movie productions and game development to digital advertising and content creation on social media platforms.

Recent advances in video motion transfer increasingly leverage large, foundational generative video models. These models typically employ the DiT architecture within a denoising diffusion loop[1]. They are not only capable of synthesizing high-quality videos from noise, but can also be conditioned with text or image prompts to control the video style and content. A variety of motion transfer approaches have emerged that leverage these powerful visual priors, in either training-based and training-free fashion. Training-based methods (*e.g.*, MotionDirector (Zhao et al., 2023b), MOFT (Zhang et al., 2023), DeT (Shi et al., 2025)) extract the motion patterns of a specific reference video by fine-tuning the parameters of the diffusion backbone. For example, MotionDirector (Zhao et al., 2023b) and DreamMotion (Jeong et al., 2024a) adopt dual-path versions of low-rank adaptation (Hu et al., 2022) to disentangle the representations of motion and appearance in the diffusion DiT. Although they are capable of generating videos whose motion follows the reference, they suffer from practical limitations: overfitting to every new reference video is time-consuming (*e.g.*, up to 2 hours on an A100 GPU) and therefore unsuitable for open-domain and real-time settings.

To achieve efficient and generally applicable motion transfer, attention has shifted to training-free frameworks (Pondaven et al., 2025a; Xiao et al., 2024; Yatim et al., 2024b). They obviate the need for per-video fine-tuning and thus enable significantly faster synthesis (*e.g.*, ≈10 minutes on an A100 GPU). The training-free approach also exploits the gradual, iterative denoising process of contemporary video foundation models: The reference video is first inverted into the embedding space of the DiT to extract features that encode the motion. Then the output video is synthesized by denoising diffusion, guided by both a text prompt *and* the gradient between the motion embeddings of the source and target video.

Our work is motivated by the observation that, in existing implementations of this pipeline, both the extraction of motion embedding from DiT backbone and the computation of motion gradients introduce considerable redundancy. Rather elementary properties of videos, and of the associated generative process, suggest that the computational cost of training-free motion transfer can be reduced considerably. *(i) **Motion redundancy***: To extract the motion embeddings from latents (Yatim et al., 2024a) or attention maps (Pondaven et al., 2025a) in the inversion stage, it is not necessary to calculate pairwise similarities between all tokens of consecutive frames. Frame-to-frame motion has limited magnitude and is locally smooth, hence motion features can be computed more efficiently, see Fig. 2(a). *(ii) **Gradient redundancy***: In the denoising stage, there is no need to recalculate all gradients at each timestep. We find that motion transfer is a case of "*stable gradient optimization*". Motivated by the idea of deterministic sampling to upgrade DDPM (Ho et al., 2020) to DDIM (Song et al., 2020), we examine the gradient updates in consecutive optimization steps and observe that they tend to be similar, see Fig. 2(b). Consequently, gradients can be reused over multiple iterations.

Based on these observations, FastVMT makes two contributions to achieve efficient motion transfer.

(1) Instead of extracting motion embeddings token by token, as in DiTFlow (Pondaven et al., 2025b), we design a sliding-window strategy that operates on downsampled attention maps and an associated corresponding window loss, to perform a more reliable and more efficient local search for motion correspondence.

(2) We address gradient redundancy with a step-skipping gradient computation. Gradients are recalculated only at selected iteration steps, between those steps, the most recent values are reused so as to reduce the total number of gradient calculations and amortize them better.

These two tricks enable high-fidelity video generation with camera trajectories and/or object motions according to the source video, see Fig. 1. Extensive experiments and user studies confirm that

---

[1]In this paper, the term "diffusion" includes flow-based interpolants (Lipman et al., 2022; Liu et al., 2022).

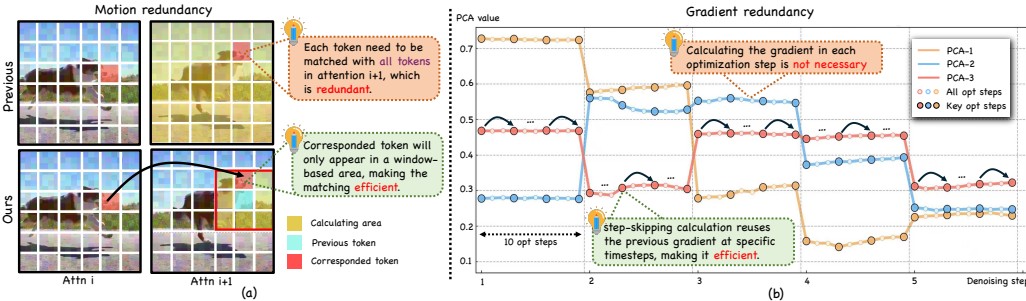

**Figure 2: Motivation of our method**. Training-free video motion transfer can benefit from redundancies, both at the level of the DiT architecture and of the iterative diffusion process. *(a) Motion redundancy*: Video motion is small and locally consistent, so a motion token in one frame will only ever match tokens in the next frame within a local neighborhood. *(b) Gradient redundancy*: Gradient updates in consecutive optimization steps are mostly similar (visualized here with PCA). There is no need to recompute them at every single step.

FastVMT achieves state-of-the-art performance both qualitatively and quantitatively, with up to **14.91×** lower latency. Furthermore, FastVMT delivers a **3.43×** speedup with minimal performance degradation, preserving near-lossless quality across various evaluation metrics when compared to the original training-free video motion transfer pipeline.

## 2 RELATED WORK

**Text-to-video generation.** Text-to-video generation aims to synthesize realistic videos by precisely matching both the visual content and motion dynamics described in the input prompt. Previous works (Long et al., 2025; Shen et al., 2025; Chen et al., 2025c; Ma et al., 2022; Feng et al., 2025; Chen et al., 2024; Guo et al., 2023; Wang et al., 2023; Ma et al., 2024b; 2026; 2025e;d;a; 2024d; 2025b;c; Qiu et al., 2025c;b; Xiong et al., 2025; Yang et al., 2024a) introduce temporal modules in UNet architectures to generate coherent videos. To generate complex video motion, the advancement of Diffusion Transformer-based methods for text-to-video generation exhibits superior performance in both spatial quality and temporal consistency. These models (Liu et al., 2024; Yang et al., 2024b; Xu et al., 2024; Kong et al., 2024; Wang et al., 2025a) demonstrate the power of scaling transformers to produce highly realistic video clips from detailed prompts, unlocking potential for diverse downstream video generation tasks.

**Video motion transfer.** Motion transfer focuses on generating novel videos while transferring motion from reference videos, differing from video-to-video translation (Zhao et al., 2023a; Ma et al., 2025d; Liu et al., 2023) by decoupling spatial appearance and temporal motion. Early approaches rely on explicit control signals such as poses (Ma et al., 2024a; Zhao et al., 2023a), depths (Gen, 2023; Xing et al., 2024), and bounding boxes (Wang et al., 2024b). Training-based methods (Zhao et al., 2023b; Jeong et al., 2024a; Ren et al., 2024) employ spatial-temporal decoupled attention mechanisms by a dual-path LoRA architecture. Recent works (Ren et al., 2024; Wu et al., 2024) improving motion-appearance disentanglement, though they remain time-consuming and non-reusable. Training-free methods (Hu et al., 2024; Pondaven et al., 2025a; Yesiltepe et al., 2024; Ling et al., 2024; Xiao et al., 2024) extract motion embeddings during inference, with DiTFlow (Pondaven et al., 2025a) proposing attention motion flow optimization. However, existing methods suffer from computational redundancy in both the architectural and diffusion process perspectives. In contrast, we first analyze the redundancy in training-free motion transfer and design the sliding-window motion extraction and step-skipping optimization to improve efficiency.

## 3 METHOD

Given an input video $\mathcal{I} = [\boldsymbol{I}^1, ..., \boldsymbol{I}^n]$, and the prompt $\mathcal{P}$ describing the target video content, we aim to design an efficient training-free framework to generate a novel video $\mathcal{J} = [\boldsymbol{J}^1, ..., \boldsymbol{J}^n]$ following the input prompt $\mathcal{P}$, while preserving the same camera pose changes and object motion.

To achieve this, we propose FastVMT, an efficient framework using DiT-based video generative model (Wang et al., 2025a) to transfer motion efficiently. The pipeline of our method is shown in Fig. 4. We first analyze the existing redundancy in previous works and introduce our motivation in Sec. 3.1. The sliding-window motion extraction strategy is present in Sec. 3.2. To improve the motion consistency, we design the corresponding window loss in Sec. 3.3. Finally, in Sec. 3.4, we propose the step-skipping gradient optimization to ensure gradient efficiency.

## 3.1 MOTIVATION

We summarize the two observed redundancies of state-of-the-art approaches in the training-free video motion transfer task and propose the modules to address them.

**Motion redundancy.** In the inversion stage, existing training-free video motion transfer approaches (Pondaven et al., 2025a; Xiao et al., 2024; Yatim et al., 2024b) utilize the global token similarity to obtain the reference motion flow. Specifically, for every optimization step, each token requires calculating the similarity with all tokens in the next attention map. However, we note that every motion token will only correspond with a token in nearby regions in the next attention map. As shown in Fig. 2(a), the corresponding token in the dog's nose would only appear around nearby regions rather than on the road. Therefore, such a property about temporal consistency makes it unreasonable to extract the motion flow by calculating token-by-token similarity globally. To address this, we introduce the sliding-window motion extraction strategy. Only the regional tokens are calculated for efficient motion extraction. Meanwhile, such a design enables correcting the mismatch during the motion extraction, as shown in Fig. 5, ensuring the motion consistency of generated results.

**Gradient redundancy.** During the optimization process of training-free motion transfer methods, a significant computational bottleneck emerges from the repetitive gradient calculations performed at each inner optimization step. Specifically, for every denoising timestep, the optimization loop performs gradient computation across all inner optimization steps to update the latent representation. However, we observe that the gradient updates exhibit high similarity across consecutive optimization steps within the same denoising timestep. As shown in Fig. 2(b), the PCA analysis reveals that gradient patterns remain relatively stable across adjacent optimization steps. Therefore, such "*stable gradient optimization*" makes it unnecessary to compute gradients at every optimization step. To address this, we introduce the step-skipping gradient optimization strategy. Only specific optimization steps require gradient computation, while intermediate steps reuse cached gradients for efficient optimization (in Fig. 3).

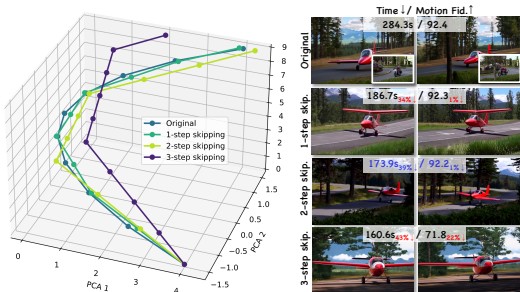

Figure 3: **Illustration of step-skipping gradient optimization**. We observe that skipping some steps in the gradient optimization step does not degrade the motion transfer performance. When we increase the skipping step, the optimization trajectory is similar until 3-step skipping.

## 3.2 EFFICIENT ATTENTION WINDOW

**Attention acquisition.** We leverage the inherent attention mechanism within video Diffusion Transformers (DiTs) to extract fine-grained motion patterns, based on the premise that correlated content across video frames is naturally captured by the self-attention layer's query-key interactions.

Given an input video $\mathcal{I} = [\boldsymbol{I}^1, ..., \boldsymbol{I}^n]$, and the prompt $\mathcal{P}$ of target video content, we utilize the 3D VAE encoder (Wang et al., 2025a) to obtain its latent representation $z_{ref} = \mathcal{E}(x_{ref})$. To obtain a clean motion signal, this latent is passed through a specific DiT block at a low denoising step, typically $t = 0$. For our tile-based approach, we first partition the spatial dimensions into tiles of size $(t_h, t_w)$. For each tile, we select a representative query at the tile center and compute its attention with all keys in the target frame. For any pair of frames $(i, j)$ in the video, the representative cross-frame

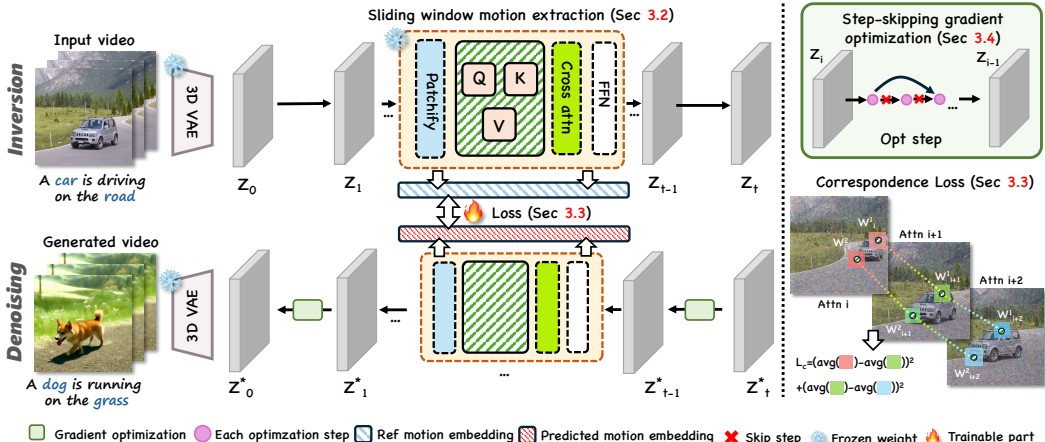

Figure 4: **Overview of our method**. *Left*: Given a reference video, we first leverage the sliding window to extract motion embedding from attention during the inversion stage. At the denoising stage, we calculate the total loss and leverage the step-skipping gradient optimization to guide the video generation. *Right:* The Step-skipping gradient optimization is proposed to improve gradient redundancy. Additionally, we introduce the corresponding-window loss to boost the motion consistency of generated videos.

attention map $\mathbf{A}_{ij}^{\text{rep}}$ is computed as:

$$\mathbf{A}_{ij}^{\text{rep}} = \text{softmax}\left(\frac{\mathbf{Q}_{\text{rep}}^{(i)}(\mathbf{K}^{(j)})^T}{\sqrt{D_h}} \cdot \tau\right) \in \mathbb{R}^{N_{\text{tiles}} \times S} \tag{1}$$

where $N_{\text{tiles}} = \frac{H}{t_h} \times \frac{W}{t_w}$ is the number of tiles, $S = H \times W$ is the spatial token length, and $\tau$ is the temperature parameter. From this representative attention map, we estimate the window center for each tile as:

$$\mathbf{c}_{uv}^{(ij)} = \sum_{s=1}^{S} \mathbf{A}_{ij}^{\text{rep}}[s] \cdot \text{pos}(s) \tag{2}$$

where $\text{pos}(s)$ denotes the spatial position of token $s$. This estimated center guides the subsequent window-constrained Attention Motion Flow (AMF) computation, enabling efficient motion extraction while maintaining spatial precision.

**Sliding-window motion extraction.** To enhance the computational efficiency and precision of AMF extraction, we propose a novel sliding window strategy that mitigates the redundant computations inherent in prior methods. Our approach leverages the observation that long-range query-key interactions in self-attention layers yield diminished motion information, and the most relevant keys for an object are typically confined to a local spatial window due to finite motion speeds.

We extract AMF from query $\mathcal{Q}$ and key $\mathcal{K}$, both of shape $(N, H, W, D)$, where $H$ and $W$ denote the height and width of the latent representation, and $N$ is the number of frames. Here, $\mathcal{Q} = \{q_1, \ldots, q_N\}$, with $q_i, i \in \{1, \ldots, N\}$ representing the query tensor for a specific frame, and $\mathcal{K}$ follows a similar notation. Unlike prior methods that compute AMF across all $q$-$k$ pairs while attending to the entire spatial dimension, our approach employs a sliding window to constrain computations both temporally and spatially:

$$\mathcal{T}_{\text{window}}(q_i) = \{q_j : j \in [i, \min(i + s_f, N)]\}, \quad \mathcal{S}_{\text{window}}(k_{h,w}) = \{k_{h',w'} : (h', w') \in \mathcal{W}_{h,w}^l\} \tag{3}$$

where $s_f$ represents the temporal span and $\mathcal{W}_{h,w}^l$ denotes a spatial window of size $l \times l$ centered at position $(h, w)$. To determine the optimal window center, we partition each frame into spatial blocks and select representative queries. The window center for each block is computed as:

$$\mathbf{c}_{\text{block}}^{(ij)} = \mathbf{P}_{\text{block}} + \text{argmax}_{(h,w)}\left(\mathbf{Q}_{\text{rep}}^{(i)} \cdot (\mathbf{K}^{(j)})^T\right)_{h,w} \tag{4}$$

where $\mathbf{P}_{\text{block}}$ is the block center position and the argmax operation yields the displacement vector from representative query-key interactions.

Our approach significantly enhances efficiency. Temporally, it reduces the time complexity from $\mathcal{O}(F^2)$ to $\mathcal{O}(F)$, where $F$ is the number of frames, enabling scalable video generation. Spatially, by constraining computations to a local window containing the most relevant keys, we eliminate redundant calculations, thereby achieving precise AMF extraction with minimal quality loss.

### 3.3 Corresponding-window Loss

Motivated by the observation that motion information is predominantly captured by closely adjacent query-key pairs, we design a weighted AMF loss and a corresponding-window loss to enhance motion transfer accuracy with temporal stability. The weighted AMF loss aligns the motion patterns between reference and generated videos by computing the $L_2$ distance between their respective displacement matrices, which is formulated as:

$$\mathcal{L}_{\text{AMF}} = \frac{1}{|\mathcal{F}|} \sum_{(i,j) \in \mathcal{F}} w_{|j-i|} \cdot \|\Delta_{ij}^{\text{ref}} - \Delta_{ij}^{\text{gen}}\|_2^2 \tag{5}$$

where $\mathcal{F}$ represents all frame pairs within the temporal span $s_f$, and the weights are defined as:

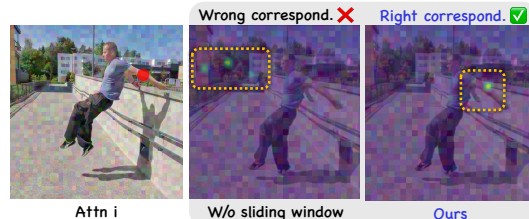

Figure 5: **Illustration of attention motion flow extraction with sliding window**. Without the sliding window, attention tokens are prone to incorrect correspondences (middle). Incorporating a sliding window improves alignment, leading to better motion consistency(right).

$$w_d = \begin{cases} 1.0 - \alpha \cdot \frac{d-1}{s_f - 1} & \text{if } d \leq s_f \\ 0 & \text{otherwise} \end{cases} \tag{6}$$

where $\alpha$ is set as 0.2 to provide linear decay, and $d = |j - i|$ represents the frame distance.

To enhance temporal consistency, we introduce a corresponding-window loss that penalizes inconsistencies in key representations across adjacent frames within the sliding windows:

$$\mathcal{L}_{\text{window}} = \frac{1}{F} \sum_{i=0}^{F-1} \frac{1}{P} \sum_{p=1}^{P} \frac{1}{N_i - 1} \sum_{t=1}^{N_i - 1} \left\| \bar{K}_{i \to j_{t+1}}^{(p)} - \bar{K}_{i \to j_t}^{(p)} \right\|_2^2, \tag{7}$$

where $\bar{K}_{i \to j}^{(p)}$ denotes the mean key representation within the sliding window $W_{i \to j}^{(p)}$ for tile $p$ when anchoring at frame $i$ and comparing with target frame $j$.

The total loss combines both components with appropriate weighting:

$$\mathcal{L}_{\text{total}} = \lambda_{\text{AMF}} \cdot \mathcal{L}_{\text{AMF}} + \lambda_{\text{window}} \cdot \mathcal{L}_{\text{window}}, \tag{8}$$

where $\lambda_{\text{AMF}}$ is set to 5 to emphasize motion alignment, and $\lambda_{\text{track}}$ is set to 1 to balance the corresponding-window loss. This dual-component loss ensures both accurate motion transfer and temporal stability, effectively addressing motion consistency challenges in video generation.

### 3.4 Step-Skipping Gradient Optimization

Despite the computational efficiency introduced by our sliding window strategy, optimizing the latent representation remains computationally intensive due to the high cost of back propagation through multiple DiT blocks. Through empirical analysis, we observe a high degree of similarity in the gradients of the latent representation across consecutive optimization steps. Leveraging this insight, we propose an interval-based gradient reuse strategy that selectively computes gradients while maintaining optimization effectiveness.

Our step-skipping optimization operates with a fixed interval $\Delta$ during the inner optimization loop. For a total of $J$ optimization steps, gradient computation occurs only when the current step $j$ satisfies

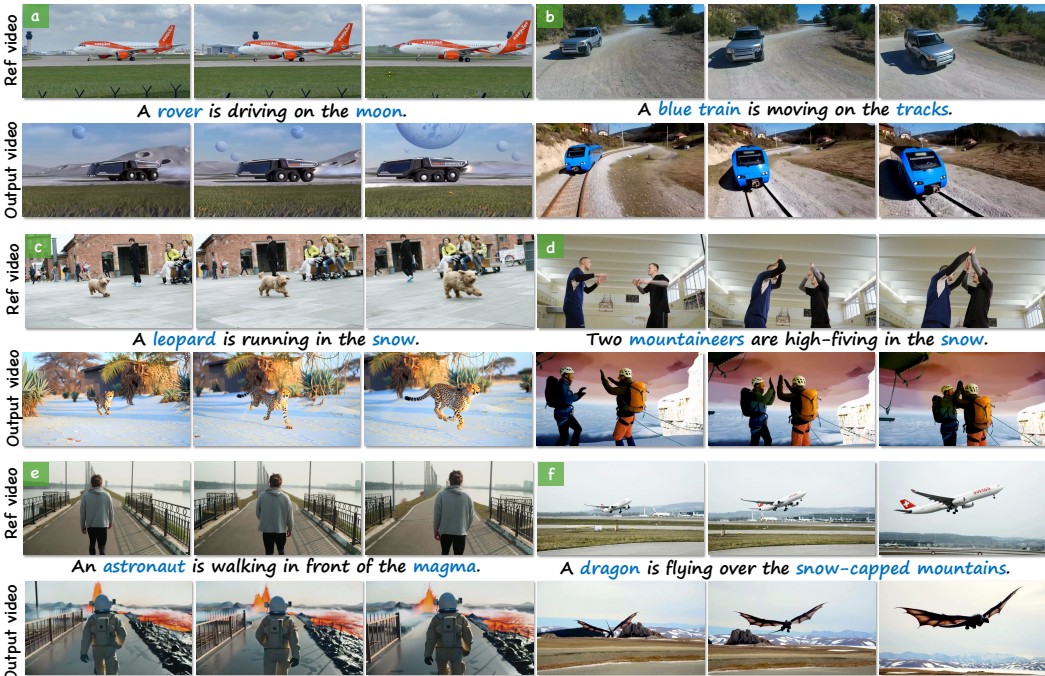

Figure 6: **Gallery of our method.** Given a reference video, our FastVMT is capable of generating high-quality video clips that faithfully preserve diverse motion patterns. More visual results can be found in Appendix C.

the condition $j \bmod \Delta = 0$, or when using the full AMF mode. The algorithm can be formalized as:

$$\mathcal{L}_j = \begin{cases} \nabla_{\mathbf{x}} \mathcal{L}_{\text{total}}(\mathbf{x}_j) & \text{if } j \bmod \Delta = 0 \text{ or mode} = \text{AMF} \\ \mathbf{x}_j \cdot \mathbf{g}_{\text{cached}} & \text{otherwise} \end{cases} \quad (9)$$

where $\mathbf{g}_{\text{cached}}$ represents the gradient from the most recent computation step. This strategy reduces gradient computations from $J$ to approximately $\lceil J/\Delta \rceil$ per guidance step, achieving a theoretical speedup of $\Delta/\lceil J/\Delta \rceil \times$ in the optimization phase. The cached gradient $\mathbf{g}_{\text{cached}}$ is updated after each actual gradient computation:

$$\mathbf{g}_{\text{cached}} = \mathbf{g}_j \text{ when } j \bmod \Delta = 0 \quad (10)$$

This approach significantly reduces computational overhead while maintaining motion transfer quality, as the gradient similarity across consecutive steps ensures that cached gradients remain effective for optimization guidance.

## 4 EXPERIMENTS

### 4.1 IMPLEMENTATION DETAILS

In our experiment, we employ the open-sourced video generation model WAN-2.1 (Wan et al., 2025) as the base text-to-video generation model. The denoising steps are employed for 50 for all experiments. Unless stated, the output resolution is $480 \times 832$ with $F = 81$ frames (internally rounded to $4k+1$). Latents are initialized as Gaussian noise of shape $\left(1, 16, \frac{F-1}{4} + 1, \frac{H}{8}, \frac{W}{8}\right)$. Latent tiling is enabled with `tile_size` $=(30, 52)$ and `tile_stride` $=(15, 26)$ in VAE space; this yields a per-frame token grid of $h = \frac{H}{8}$ by $w = \frac{W}{8}$ for the DiT. During motion transfer, as Pondaven et al. (2024), we enable our sliding-window based AMF guidance at the first 20% outer denoising steps; each guided step runs a 10-step latent-only inner optimization with AdamW and a linear learning-rate decay $0.003 \rightarrow 0.002$. At each guided diffusion step $t$, we form a reference latent by adding step-consistent noise to cached clean latents and perform a forward pass with null text to extract queries/keys from the 15th DiT block. More details can be found in the Appendix 5.

## 4.2 COMPARISON WITH BASELINES

**Qualitative comparison.** We compare our approach with the state-of-the-art video motion transfer methods visually: MOFT (Xiao et al., 2024), MotionInversion (Wang et al., 2024a), Motion-Clone (Ling et al., 2024), SMM (Yatim et al., 2024b), MotionDirector (Zhao et al., 2023b), DiT-Flow (Pondaven et al., 2024), and DeT (Shi et al., 2025). For fair comparison, we adapt the Wan-2.1 as the same backbone. Our experimental results demonstrate that FastVMT achieves superior performance and greater versatility across a wide range of motion transfer scenarios. As illustrated in Fig. 8, these works (Xiao et al., 2024; Yatim et al., 2024b; Pondaven et al., 2025a; Shi et al., 2025) have the challenge of handling complicated interaction motion. In contrast, our method enables generating videos with aligned movement patterns, preserving the spatial relationships between moving subjects.

**Quantitative comparison.** We compare our method with the state-of-the-art video motion transfer on on 50 high-quality videos selected from the DAVIS dataset (Perazzi et al., 2016). For fair comparison, we employ Wan-2.1 as the same backbone. Previous works are constrained by the limited video length, with evaluations conducted using only 32 frames at a resolution of $830 \times 480$. In this context, we classify the state-of-the-art (SOTA) methods into two categories: training-free and tuning-based, based on whether they leverage spatial/temporal LoRA for optimizing complex motion patterns. (a) **Time**: We record the total time required for completing the motion transfer process, including any inference-time optimization. Leveraging proposed sliding-window motion extraction and step-skipping gradient optimization, FastVMT is the fastest method. Its runtime is faster than training-free methods, while delivering better performance. (b) **Motion Fidelity:** As in Yatim et al. (2024b), we use motion fidelity to assess the similarity of tracklets between reference and generated videos. (c) **Temporal Consistency:** We measure frame-to-frame coherence by calculating the average feature similarity of consecutive video frames using CLIP (Radford et al., 2021). (d) **Text Similarity:** CLIP is used to extract features from the target video, and the average cosine similarity between the input prompt and video frames is computed. (f) **User Study:** To account for the limitations of automatic metrics in capturing real-world preferences, we conducted a user study with 20 volunteers. They ranked methods based on motion preservation, appearance diversity, text alignment, and overall quality, using a 1 (best) to 8 (worst) scale. The average rank per method (lower ranks are better) is presented in Appendix.5. Our method outperforms others in both automated metrics and user preferences.

In addition, we collect 40 real-world videos and 40 high-quality generated videos by advanced text-to-video generative models (Kong et al., 2024; Wang et al., 2025b). For each video, we generate 5 different prompts. Four metrics in VBench (Huang et al., 2023) are employed for a more accurate evaluation (in Tab. 2). (1) **Subject Consistency:** We assess whether the identity of the subject is preserved across frames. (2) **Motion Smoothness:** The metric evaluates interframe continuity using learned motion priors. (3) **Aesthetic Quality** uses a LAION-trained aesthetic predictor to score visual appeal. (4) **Background Consistency:** We evaluate the coherence of the background. Our proposed method significantly outperforms all baseline approaches across every video quality metric, thereby showcasing the state-of-the-art performance in novel video.

Table 1: **Quantitative ablation**. **Red** and **Blue** denote best, 2nd.

| Method | Text Sim.↑ | Motion Fid.↑ | Temp. Cons.↑ | Time(s)↓ |
|---|---|---|---|---|
| w/o Sliding Wind. | 0.2352 | 0.6912 | 0.9654 | 227 |
| w/o Cor. Loss | **0.2345** | 0.5942 | 0.9762 | **183** |
| w/o Step Skip. | 0.2317 | **0.7044** | **0.9881** | 302 |
| Ours | **0.2422** | **0.7471** | **0.9865** | **184** |

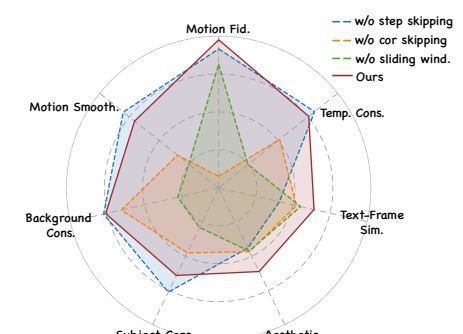

Figure 7: **Quantitative ablation comparison on Vbench metrics**. We select the seven metrics to evaluate the effectiveness of the proposed strategy.

Table 2: **Comparison with state-of-the-art video motion transfer methods**. Red and Blue denote the best and second best results, respectively. User study scores are reported in Appendix 5.

| Method | Quantitative Metrics | | | | Vbench Metrics | | | |
|---|---|---|---|---|---|---|---|---|
| | Text Sim.↑ | Motion Fid.↑ | Temp. Cons.↑ | Time (s)↓ | Sub. Cons.↑ | Back. Cons.↑ | Aes. Qual.↑ | Motion Smooth.↑ |
| **Training-Based Methods** | | | | | | | | |
| MotionInversion (Jeong et al., 2024b) | 0.2388 | 0.6515 | 0.9605 | 632.41 | 0.9339 | 0.9372 | 0.4062 | 0.9532 |
| MotionDirector (Zhao et al., 2023b) | 0.2336 | 0.4524 | 0.9531 | 806.64 | 0.9173 | 0.9379 | 0.3443 | 0.9633 |
| DeT (Shi et al., 2025) | 0.2187 | 0.6116 | 0.9818 | 2745.60 | 0.9787 | 0.9654 | 0.3559 | 0.9598 |
| **Training-Free Methods** | | | | | | | | |
| MOFT (Xiao et al., 2024) | 0.2297 | 0.6511 | 0.9797 | 595.81 | 0.9593 | 0.9413 | 0.4581 | 0.9716 |
| MotionClone (Ling et al., 2024) | 0.2304 | 0.7315 | 0.9722 | 397.05 | 0.9601 | 0.9545 | 0.4615 | 0.9616 |
| SMM (Yatim et al., 2024b) | 0.2374 | 0.7353 | 0.9366 | 809.70 | 0.8907 | 0.9352 | 0.5770 | 0.9702 |
| DiTFlow (Pondaven et al., 2025a) | 0.2091 | 0.4062 | 0.9822 | 626.83 | 0.9557 | 0.9678 | 0.5310 | 0.9801 |
| Ours | 0.2422 | 0.7471 | 0.9865 | 184.18 | 0.9809 | 0.9684 | 0.5778 | 0.9891 |

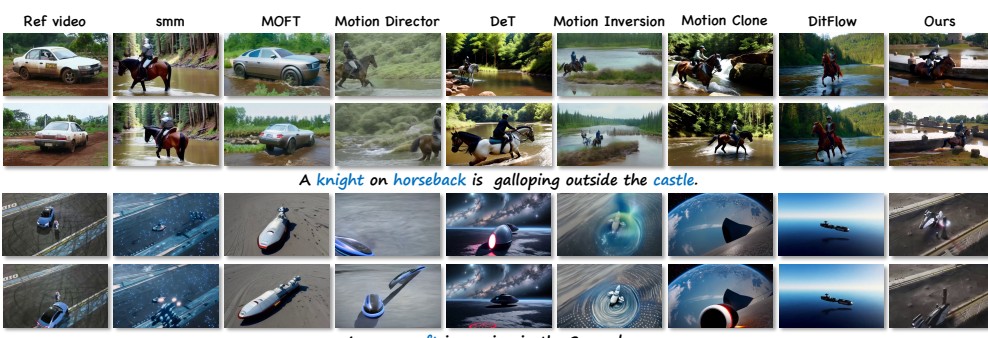

A knight on horseback is galloping outside the castle.

A spacecraft is moving in the Space base.

Figure 8: **Qualitative comparison with baselines.** We perform the visual comparison with various baselines using various kinds of motions. Our method obtains better performance in various motions. More visual results can be found in Appendix C.

### 4.3 ABLATION STUDY

**Effectiveness of sliding-window based motion extraction.** As shown in Tab. 1 and Fig. 7, removing the sliding window mechanism results in performance degradation across multiple metrics and increased computational overhead and inference time. In Fig. 10, we present the visual results without sliding windows. It is clear to observe a light reduction in motion fidelity and temporal consistency, confirming that our approach effectively balances computational efficiency with motion transfer quality. Additionally, we also show the visual comparison of attention motion extraction in various attention layers in DiT (see Fig. 9). The motion extraction is more accurate in the middle layer of DiT. The quantitative ablation about it is provided in Appendix 5.

**Effectiveness of corresponding-window loss.** Tab. 1 and Fig. 7 reveal that excluding the corresponding-window loss leads to substantial degradation in motion fidelity, highlighting its essential role in maintaining accurate motion transfer. As shown in Fig. 10, equipping with this loss function effectively constrains temporal inconsistencies to ensure robust motion alignment, while introducing minimal computational overhead (less than 1% increase in processing time), thus preserving both accuracy and efficiency.

**Effectiveness of step skipping gradient upgrading.** The step-skipping strategy significantly reduces computational time while preserving video generation quality. As demonstrated in Tab. 1 and Fig. 7, this optimization achieves substantial time savings with negligible impact on motion fidelity and temporal consistency, validating the effectiveness of gradient reuse in our framework.

## 5 CONCLUSION

In this work, we introduced FastVMT, a training-free video motion transfer framework that explicitly addresses *motion redundancy* in diffusion transformer architectures and *gradient redundancy* along the diffusion trajectory. To eliminate the motion redundancy, we propose the sliding-window strategy associated with corresponding window loss to achieve a more reliable and more efficient local search for motion correspondence. To migrate gradient redundancy, We introduce a step-skipping gradient

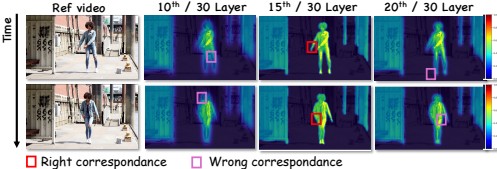

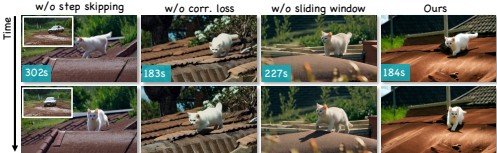

Figure 9: **Illustration of token correspondence performance in various attention layers of DiT**. We extract attention correspondences from different DiT layers. The middle attention layers exhibit stronger token correspondence performance.

Figure 10: **Qualitative ablation of the proposed modules.** The reference video is shown at the top-left of the first column. The inference time is shown at the bottom-left of the image. The prompt is "A white cat is running on the ground".

computation to ensure computational efficiency. By incorporating the proposed strategies, our method achieves a **3.43×** average speedup without compromising either visual fidelity or temporal consistency. We believe this line of work opens new opportunities for building more efficient and practical generative video motion transfer.

## ACKNOWLEDGEMENT

This research was supported by the Shanghai Science and Technology Program (Grant No. 25ZR1402278). Part of the compute is supported by the SwissAI Compute Grant a144 and a154.

## REPRODUCIBILITY STATEMENT

All quantitative tables, qualitative images, and video results in this work are reproducible and correspond to raw model outputs without manual editing or post-hoc alteration, except for minimal format conversion and compression. After the review process, we will release a partial public repository to support reproduction, including inference scripts, example data, and example videos. The datasets, configurations, and procedures used for training and evaluation are documented in Section 4.1 and Appendix 5. We will also provide fixed configuration files and random seeds so that independent runs can reproduce the visual results within expected stochastic variation.

## ETHICS STATEMENT

Our work studies motion-transfer video editing. The proposed dataset contains videos of people, vehicles, and landscape camera motions. To mitigate representational bias in demonstrations, we curated and displayed examples spanning different races, genders, and styles in the main text and appendix. All illustrative videos shown in this paper are sourced from publicly available web content; we respect the original licenses and terms of service and use the content solely for research purposes. We will not publicly release the dataset prior to completing the insertion of AI-generated watermarks and an ethics/content-safety audit. We explicitly prohibit harmful or deceptive uses of our methods and data, including deepfake attacks and other malicious generative behaviors. When any portion of our code is made public, we will enforce visible and/or machine-detectable watermarking during inference to help deter misuse. Any future releases will be accompanied by usage terms that forbid impersonation, harassment, or other malicious applications, and we will remove or restrict content that raises privacy, legal, or safety concerns.

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

APPENDIX

## A   DYNAMIC VIDEOS

We show more video motion transfer results produced by our method in an MP4 file, which can be found in the file: `demo.mp4`.

## B   MORE DETAILS ABOUT IMPLEMENTATION

### B.1   IMPLEMENTATION DETAILS

Our sliding-window based AMF uses a tile grid of $3 \times 4$ tokens, a temporal span $s_f = 5$, a local search window $l = 21$ (half-width 10), and temperature $\tau = 1.0$. The AMF loss between the reference and current sample is a weighted squared $\ell_2$ over temporal offsets with linear weights $1.0 \rightarrow 0.8$ across offsets, normalized by the number of offsets and tokens; gradients update only the latent $x$. To reduce cost, within the inner loop we recompute Q/K and tile-AMF every `interval`$= 3$ steps and reuse the cached gradient on intermediate steps.

### B.2   RELATED WORK

**Text-to-video generation.** Text-to-video (T2V) generation targets to synthesize realistic videos by precisely matching both the visual content and the motion dynamics described in the prompt. Early approaches (Zhu et al., 2025; Chen et al., 2024; 2023; Guo et al., 2023; Zhang et al., 2025g;d;b;a;e;c;f; Qiu et al., 2025d;a; Wang et al., 2023; Xiong et al., 2025; Yang et al., 2024a) introduce temporal modules in UNet architectures to generate coherent videos, while diffusion-based models (Guo

---

**Algorithm 1** FastVMT Algorithm

---

**Require:** $\mathbf{z}_{\text{ref}}$: reference video latent, $\mathbf{z}_{\text{gen}}$: generating latent
**Ensure:** Optimized $\mathbf{z}_{\text{gen}}$

1: **function** OPTIMIZATION($\mathbf{z}_{\text{ref}}, \mathbf{z}_{\text{gen}}$)
2:     Align noise level: $\mathbf{z}'_{\text{ref}} \leftarrow$ MatchNoise($\mathbf{z}_{\text{ref}}, \mathbf{z}_{\text{gen}}$)
3:     Do inference: DiT($\mathbf{z}'_{\text{ref}}, \mathbf{z}_{\text{gen}}$)
4:     Extract self attn features: $\mathbf{q}_{\text{gen}}, \mathbf{k}_{\text{gen}}, \mathbf{q}_{\text{ref}}, \mathbf{k}_{\text{ref}} \leftarrow$ AttnFeatures
5:     Calculate displacement matrix: $\mathcal{D} \leftarrow$ CalDisplace($q, k$)
6:     Computing loss: $\mathcal{L} \leftarrow$ LossFunc($\mathcal{D}_{\text{gen}}, \mathcal{D}_{\text{ref}}$)
7:     Backpropagate and optimize: $\mathbf{z}'_{\text{gen}} \leftarrow$ Optimization($\mathbf{z}_{\text{gen}}$)
8:     Output $\mathbf{z}'_{\text{gen}}$
9: **end function**

10: **for** $t = 1$ to $n$ **do**
11:     **if** $t < T_{\text{opt}}$ **then**
12:         $\mathbf{z}_{\text{gen}} \leftarrow$ OPTIMIZATION($\mathbf{z}_{\text{ref}}, \mathbf{z}_{\text{gen}}$)
13:     **end if**
14:     $\mathbf{z}_{\text{gen}} \leftarrow$ DENOISE($\mathbf{z}_{\text{gen}}$)
15: **end for**

---

Table 1: **User Study Comparison for State-of-the-Art Video Motion Transfer Methods**. The results show the average rank (1=best, 8=worst) for all the methods; lower is better. **Red** and **Blue** denote the best and second best results.

| Method | User Study | | | |
|---|---|---|---|---|
| | Motion Pres.↓ | Gen. Qual.↓ | Text Align.↓ | Overall↓ |
| **Training-Free Methods** | | | | |
| MOFT (Xiao et al., 2024) | 5.213 | 4.088 | 4.700 | 4.667 |
| MotionClone (Ling et al., 2024) | 5.300 | 4.688 | 5.362 | 5.117 |
| SMM (Yatim et al., 2024b) | **4.338** | 6.075 | 4.975 | 5.129 |
| DiTFlow (Pondaven et al., 2025a) | 4.713 | 3.200 | 4.088 | 4.000 |
| **Tuning-Based Methods** | | | | |
| MotionInversion (Jeong et al., 2024b) | 5.050 | 6.350 | 5.075 | 5.492 |
| MotionDirector (Zhao et al., 2023b) | 5.325 | 5.862 | 5.575 | 5.588 |
| DeT (Shi et al., 2025) | 4.350 | **3.175** | **3.825** | **3.783** |
| Ours | **1.712** | **2.562** | **2.400** | **2.225** |

et al., 2024; Zhao et al., 2023a; Zhang et al., 2025h; Ma et al., 2024c) leverage pretrained image diffusion models for temporal consistency. Recently, Diffusion Transformer-based methods exhibit superior performance, with models like Sora (Liu et al., 2024), CogVideoX (Yang et al., 2024b), EasyAnimate (Xu et al., 2024), HunyuanVideo (Kong et al., 2024), and Wan2.1 (Wang et al., 2025a) demonstrating the power of scaling transformers for high-quality video generation. However, these models face computational bottlenecks in attention mechanisms and optimization processes, particularly for iterative video editing tasks. Our work identifies task-specific redundancies in video diffusion models to enable efficient motion transfer.

## B.3 HUMAN EVALUATION

We conducted a user study via a questionnaire comprising 8 distinct input videos spanning 4 categories: camera motion, complex human motion, single object, and multiple objects. Videos were generated using our proposed method alongside other baseline approaches. The user study interface is illustrated in Figures 1 and 2. Owing to page constraints, only two generated videos are presented here.

## B.4 CORRESPONDING-WINDOW LOSS

We compute the head-averaged self-attention queries and keys from a fixed DiT block, denoted by $Q, K \in \mathbb{R}^{F \times H \times W \times D}$ for $F$ frames, an $H \times W$ spatial token grid, and channel dimension $D$. The spatial grid is partitioned into non-overlapping tiles $\{\mathcal{T}_p\}_{p=1}^{P}$ of size $(t_h, t_w)$, where $P = \frac{H}{t_h} \frac{W}{t_w}$. For

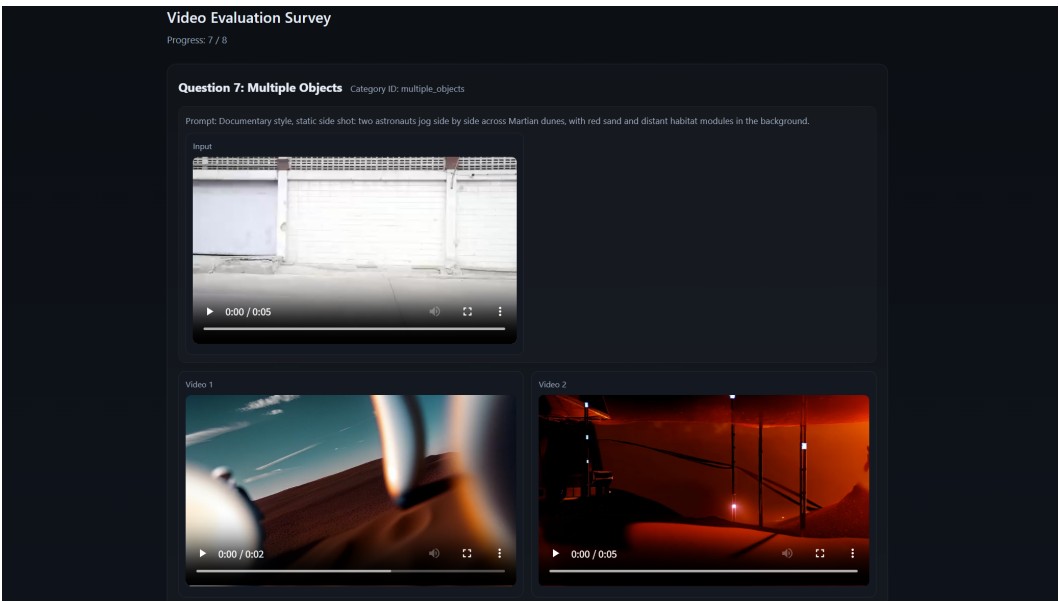

Figure 1: Input video, target prompt, and video choices as presented in the user study questionnaire. Owing to page constraints, only two videos are shown here.

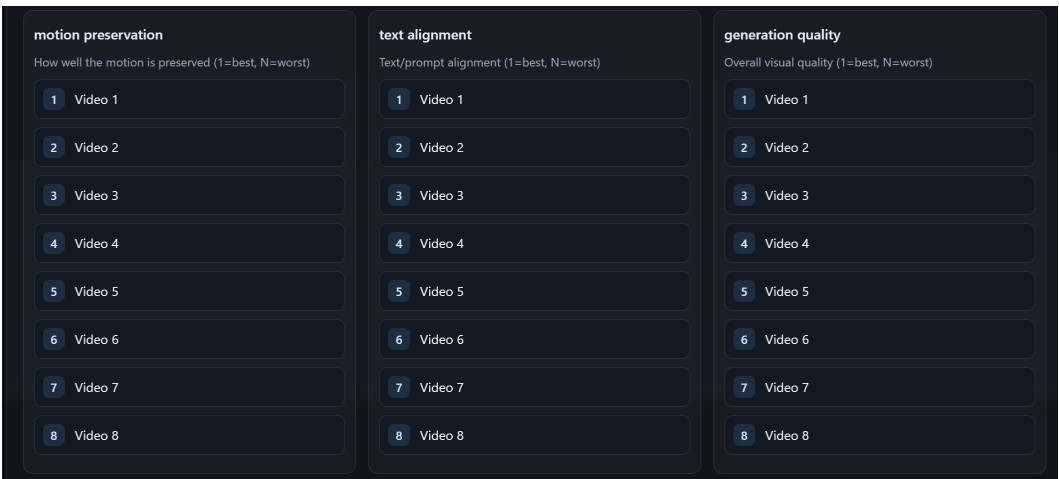

Figure 2: User study choices: Participants are prompted to rank 8 videos in descending order of preference.

an anchor frame $i \in \{0, \ldots, F-1\}$ and a temporal neighborhood $\mathcal{J}_i = \{\, i+1, \ldots, \min(i + s_f - 1, F-1)\,\}$ with $N_i = |\mathcal{J}_i|$, we define, for each tile $p$, a fixed-size window $\mathcal{W}_{i \to j}^{(p)} \subset \{1, \ldots, H\} \times \{1, \ldots, W\}$ on the target frame $j \in \mathcal{J}_i$ (windowing rule specified in the main paper). The window-averaged key feature is

$$\bar{K}_{i \to j}^{(p)} \;=\; \frac{1}{|\mathcal{W}_{i \to j}^{(p)}|} \sum_{(u,v) \in \mathcal{W}_{i \to j}^{(p)}} K_j(u,v) \;\in \mathbb{R}^D, \qquad K_j(u,v) \;=\; K[j, u, v, :].$$

Stacking these per-tile temporal features yields $K_i^{(p)} = \big[\bar{K}_{i \to j}^{(p)}\big]_{j \in \mathcal{J}_i} \in \mathbb{R}^{N_i \times D}$. The tracking loss penalizes first-order temporal variations of the window means across adjacent target frames and averages over tiles and anchors:

$$\Delta K_i^{(p)}(t) \;=\; \bar{K}_{i \to j_{t+1}}^{(p)} - \bar{K}_{i \to j_t}^{(p)}, \quad t = 1, \ldots, N_i - 1, \qquad \mathcal{L}_{\text{window}}^{(i)} \;=\; \frac{1}{P} \sum_{p=1}^{P} \frac{1}{N_i - 1} \sum_{t=1}^{N_i - 1} \left\| \Delta K_i^{(p)}(t) \right\|_2,$$

Table 2: **Ablation comparison of different attention layers (30 in total)**. **Red** and **Blue** denote the best and second best results, respectively.

| Method | Text Sim.↑ | Motion Fid.↑ | Temp. Cons.↑ | Sub. Cons.↑ | Back. Cons.↑ | Aes. Qual.↑ | Motion Smooth.↑ |
|---|---|---|---|---|---|---|---|
| 10-th layer | 0.2241 | 0.7128 | **0.9730** | **0.9610** | **0.9530** | 0.5595 | 0.9736 |
| 15th layer(Ours) | **0.2422** | **0.7471** | **0.9865** | **0.9809** | **0.9684** | **0.5778** | **0.9891** |
| 20-th layer | **0.2319** | **0.7213** | 0.9701 | 0.9549 | 0.9414 | **0.5606** | **0.9791** |

Table 3: **Ablation study about temporal span**. **Red** and **Blue** denote the best and second best results, respectively.

| Temp. Span | Sub. Cons. | Back. Cons. | Aes. Qual. | Motion Smooth. |
|---|---|---|---|---|
| span-3 | 0.9592 | 0.9461 | **0.5690** | **0.9899** |
| **span-5** | **0.9809** | **0.9684** | **0.5778** | 0.9891 |
| span-7 | **0.9711** | **0.9608** | 0.5522 | 0.9858 |

$$\mathcal{L}_{\text{window}} \;=\; \frac{1}{F} \sum_{i=0}^{F-1} \mathcal{L}_{\text{window}}^{(i)}.$$

Equivalently, writing $K_i^{(p)} \in \mathbb{R}^{N_i \times D}$ as a temporal sequence, the inner sum is the mean L2 norm of the finite differences $K_i^{(p)}[2:] - K_i^{(p)}[1:-1]$. In practice we compute $\bar{K}_{i \to j}^{(p)}$ in FP32 before reduction, and the overall guidance objective during latent optimization combines attention motion flow matching and tracking:

$$\mathcal{L} \;=\; \lambda_{\text{amf}} \, \mathcal{L}_{\text{amf}} \;+\; \lambda_{\text{window}} \, \mathcal{L}_{\text{window}},$$

with constants $\lambda_{\text{amf}} > 0$ and $\lambda_{\text{window}} > 0$.

## C  MORE COMPARSONS

In Fig. 5, we present additional comparisons to assess the performance of the proposed method. It is clear that previous works exhibit inconsistent motion. In contrast, our approach effectively resolves the issue of motion consistency.

## D  MORE RESULTS

### D.1  MORE VISUAL RESULTS

We presented more visualizations in Figure 3 and 4, where each reference video is paired with two distinct motion transferred videos. In particular, Fig. 7 presents two challenging visual cases: one featuring complex object motion and another involving complex camera motion. The first set of images shows an astronaut doing a front flip off the deck into the water. The second set of images illustrates a complex camera move, where the viewpoint rises rapidly from ground level and then pushes in for a close-up of the subject.

### D.2  MORE ABLATION RESULTS

We conduct an ablation study to evaluate the impact of selecting different attention layers for motion extraction. As illustrated in Table 2, our choice of the middle attention layer achieves the best performance in motion transfer.

We evaluated our ablation samples on Vbench metrics. The results are presented in Table 4.

### D.3  MORE QUANTITATIVE COMPARISON

We further conducted our experiment using MTBench. Our quantitative results are shown in Table **??**.

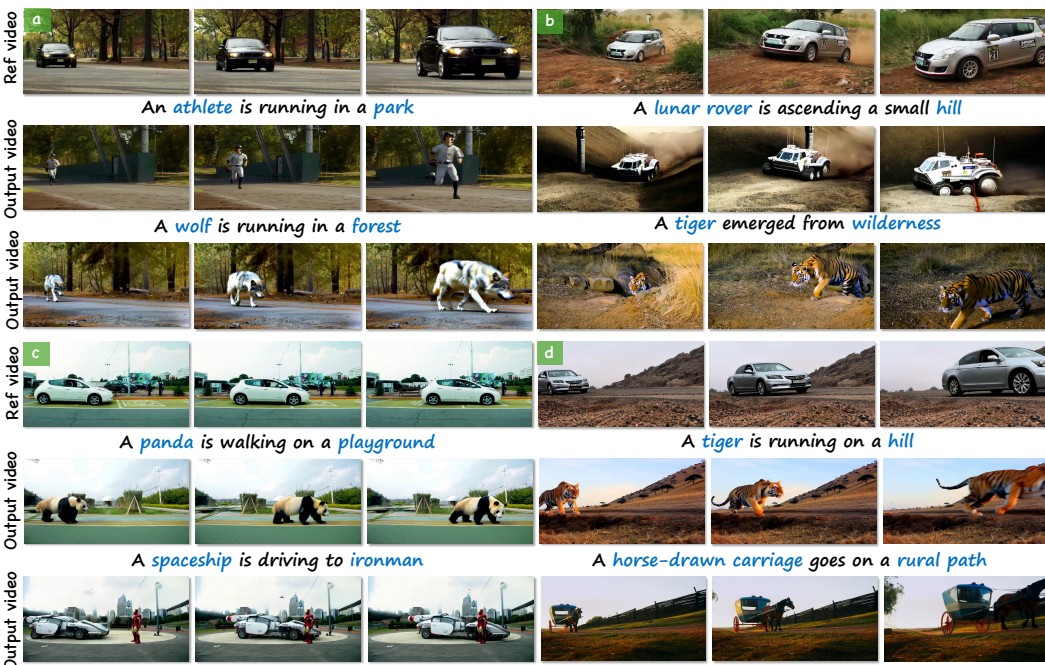

Figure 3: **More visual results.** We provide more visual results to evaluate the performance.

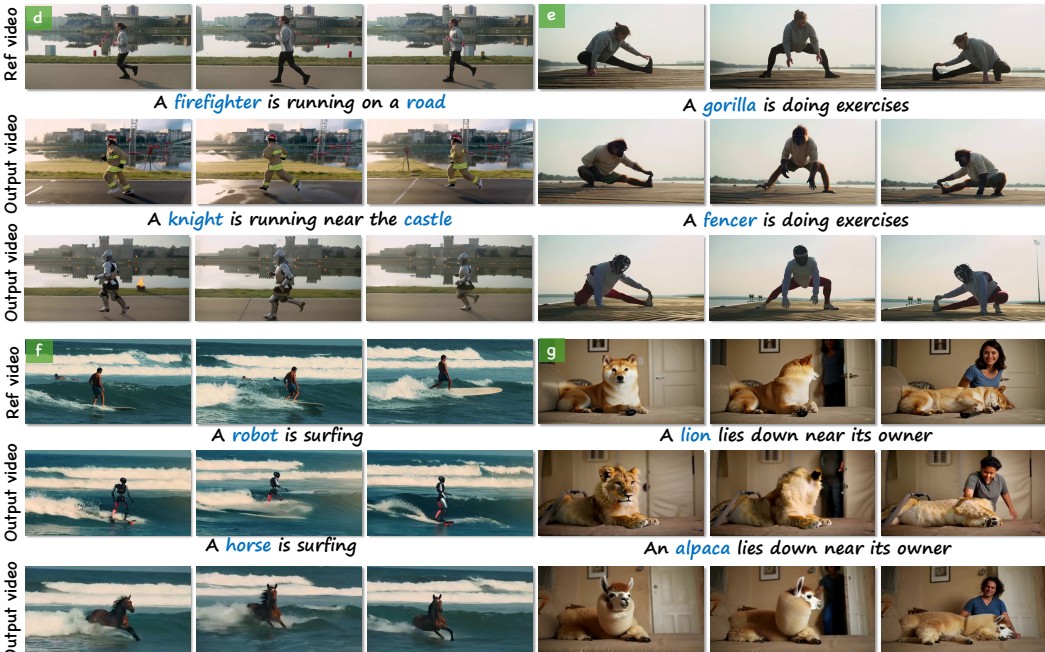

Figure 4: **More visual results.** We provide more visual results to evaluate the performance.

# E EXPERIMENT DETAILS

All experiments presented in this study were conducted utilizing NVIDIA A100-80GB GPUs for fair comparison. The reference videos were carefully curated from publicly available sources on the internet, ensuring a diverse and representative dataset for evaluation purposes.

Table 4: **Vbench metrics evaluated of the ablation samples**. Red and Blue denote the best and second best results, respectively.

| Method | Sub. Cons. | Back. Cons. | Aes. Qual. | Motion Smooth. |
|---|---|---|---|---|
| w/o Sliding Wind. | 0.9686 | 0.9437 | 0.5628 | 0.9667 |
| w/o Cor. Loss | 0.9753 | 0.9574 | **0.5629** | 0.9705 |
| w/o Step Skip. | **0.9852** | **0.9617** | 0.5623 | **0.9887** |
| Ours | **0.9809** | **0.9684** | **0.5778** | **0.9891** |

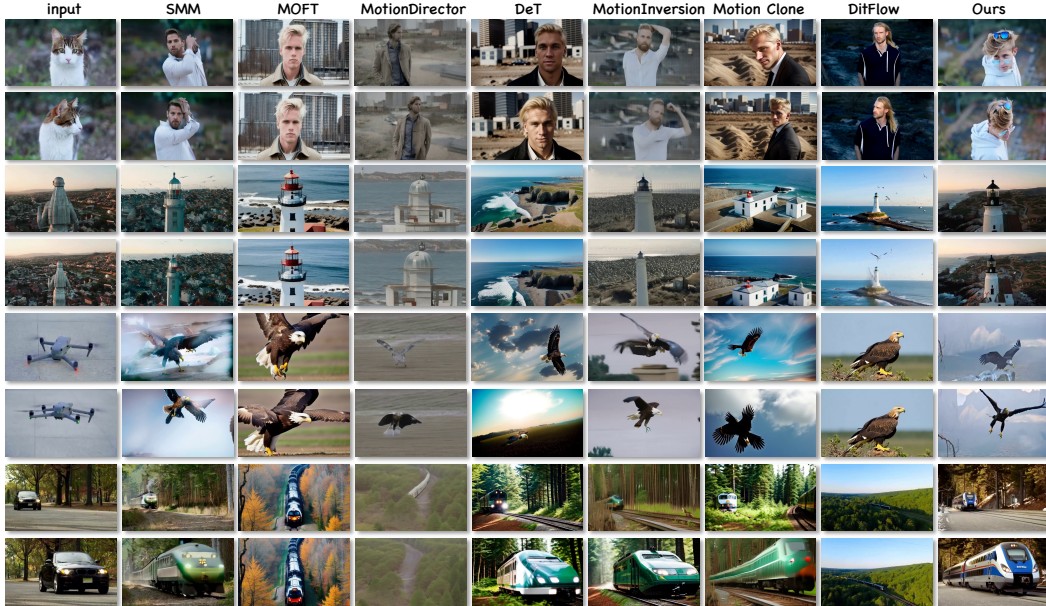

Figure 5: **More qualitative comparison with baselines.** We provide more visual comparison with various baselines using various kinds of motions. Our method demonstrates superior performance across a range of motion types.

## F    LIMITATIONS AND POTENTIAL SOCIAL IMPACT

### F.1    LIMITATION

As observed in prior work (Pondaven et al., 2025a), existing frameworks are still bounded by the capacity of the pre-trained video backbone, making it challenging to handle out-of-distribution prompts or motions. For instance, highly complex human actions (such as Thomas Flair) remain particularly difficult. When the generated video content and the conditioning prompt exhibit semantic inconsistency or conflict, the quality of motion transfer can degrade significantly, often leading to unsatisfactory or unstable results.

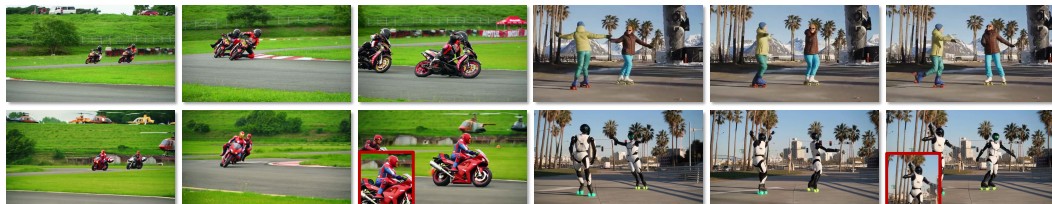

A Spider-Man and Iron man are riding motorcycles          Two cars are roller skating in an outdoor urban

Figure 6: **Failure case of proposed.** Even though our method achieves good performance, we still have a challenge when handling the occlusion. This failure can be mitigated by a powerful video diffusion model in the future.

An astronaut is doing a front flip into the water.          A detective is standing at the train station.

Figure 7: **Complex cases of proposed.** Our method also performs well in some cases with complex object motion and camera movement.

The pairwise design adopted by AMF, while beneficial for capturing motion correspondences, inevitably introduces higher memory consumption compared to prior methods. Although this overhead does not critically affect short video synthesis, it may pose practical challenges when scaling to long video generation. Looking ahead, we believe that this issue can be mitigated through systematic engineering optimizations, such as more efficient memory management strategies, model compression, or hierarchical generation schemes.

## F.2    FUTURE WORK

An important future direction is to extend our framework toward an agent-based paradigm for practical application (Lin et al., 2025a; Wang et al., 2025e;d;c; Lin et al., 2025b;c; Liu et al., 2025; Zhao et al., 2024; 2026; 2025; Song, 2022; Song et al., 2023; Song & Zhang, 2022; Qiu et al., 2024; Chen et al., 2025b;a). Rather than formulating motion transfer as a single-pass motion-conditioned generation problem, we envision a multi-agent architecture in which specialised agents collaboratively handle motion decomposition, structural alignment, identity preservation, and temporal stabilisation. Specifically, a Motion Parsing Agent could extract structured motion representations (e.g., skeletal trajectories, optical flow fields, or 3D pose sequences) from the driving signal; a Structure Alignment Agent could enforce geometric consistency between the source character and target motion manifold; an Appearance Preservation Agent could maintain identity-specific attributes under large pose deformation; and a Temporal Consistency Agent could suppress drift and flickering across frames. Through iterative reasoning and tool invocation (e.g., pose estimation, tracking, depth inference, or diffusion-based motion refinement modules), these agents could decompose complex motion transfer objectives into hierarchically organised sub-tasks and dynamically refine intermediate states. Such a formulation enables adaptive long-horizon motion synthesis, mitigates cumulative structural distortion, and improves controllability under ambiguous or partially specified motion prompts.

## F.3    POTENTIAL SOCIAL IMPACT

The potential social impact of FastVMT and efficient video motion transfer technologies is far-reaching, with applications spanning across multiple industries. In the entertainment sector, particularly in film, gaming, and digital content creation, the ability to quickly and accurately transfer motion from one video sequence to another enables faster production cycles and more dynamic storytelling, reducing costs and enhancing creativity. This could democratize high-quality video production, making it accessible to smaller studios and independent creators who previously lacked the resources to produce complex motion sequences.

In the advertising industry, FastVMT offers new opportunities for creating personalized and engaging content. Brands can easily adapt their campaigns to various demographics by transferring motion from diverse sources, ensuring relevance and resonance with their audience. Additionally, this technology could be employed for real-time video adaptation in interactive applications, further improving customer experiences.

Beyond media and entertainment, the technology also holds promise in education, remote work, and healthcare. Virtual simulations and immersive training environments could benefit from enhanced motion transfer capabilities, allowing for realistic and adaptable scenarios. This could support remote learning, telemedicine, and virtual conferences, making such interactions more engaging and effective.

Overall, FastVMT's ability to reduce computation costs and improve video synthesis efficiency can drive widespread innovation, making advanced video manipulation more accessible, affordable, and impactful across various sectors, ultimately shaping the future of digital media and interaction.

## G    THE USAGE OF LARGE LANGUAGE MODELS

In this paper, the usage of the LLM mainly falls into the following aspects:

- **Grammar checking and format optimization**: In the paragraphs of the paper, LLMs are used for grammar error checking and format checking of charts and graphs.
- **Language polishing**: The text description part of the paper uses LLMs to polish and optimize the language expression.
- All authors are responsible for the content generated by the LLMs.

