# OpenReview forum: "FastVMT: Eliminating Redundancy in Video Motion Transfer"
_ICLR.cc/2026/Conference — ICLR 2026 Poster_

### Official Review · Reviewer_RmCa · 2025-10-20

**Soundness:** 3
**Presentation:** 4
**Contribution:** 3
**Rating:** 6
**Confidence:** 5

**Summary:**

This paper introduces FastVMT, a training-free video motion transfer framework aiming at enhancing computational efficiency without sacrificing video quality or motion fidelity. To achieve this, FastVMT includes two key innovations: 1) sliding-window motion extraction which only computes attention with local spatial neighborhoods to better capture motion correspondences while eliminating unnecessary token interactions; 2) step-skipping gradient optimization that reuses gradients across optimization steps as the authors found that gradients change slowly along the diffusion trajectory. Experiments show that FastVMT yield an average 3.43× speedup and up to 14.9× lower latency compared to prior state-of-the-art methods (e.g., MotionDirector, DiTFlow, MOFT) without visible quality degradation.

**Strengths:**

- The proposed method is elegant and well-motivated. For example, both the ideas of sliding window motion extraction and step-skipping gradient optimization are lightweight but sounds effective and should work smoothly with existing video diffusion models. To introduce these two modules, the authors provide an insightful analysis (i.e, Figure 2), showing that motion is locally consistent and the gradient in consecutive steps are mostly similar.
- The efficiency improvement is significantly. Specifically, the authors demonstrate 3.43× speedup and up to 14.9× lower latency compared to the existing models, which are impressive.
- The authors provide comprehensive comparisons with the up to 7 existing models (including both training-free and finetuning-based baselines).
- The authors also provide extensive ablation studies, which clearly show the necessity of each proposed component. The results look promising.
- The paper is well written and easy to follow. The figures are well-plotted and informative which can make readers quickly understand the core ideas.

**Weaknesses:**

My first two concerns circle around the usage of sliding window attention:

- First, the usage of sliding-window attention leads to irregular and non-contiguous tiling (i.e., the attention mask is no longer a uniform square but contains irregular zero-paddings to mask out non-local tokens). This irregularity would make the model incompatible with Flash Attention that requires full or casual mask. How did the authors handle this issue? If the model does not use Flash Attention, did the authors notice a speed degradation when switching to other attention functions?
- Second, the Register Tokens paper [1] found that the transformer models tend to learn a few of register tokens which are attached by most of the tokens and used to aggregate and spread global information. However, the proposed sliding-window design restrict the receptive field of a token to its local neighborhood and further block such information exchange mechanism. This could make the model fail to handle the videos with larger dynamics or longer duration where the global information exchange is critical. However, all the video examples provided in the paper and the supplementary material are 5 seconds and only include smooth and slight motion. Could the authors provide the videos with longer sequences and larger motion to evaluate whether the proposed model can handle such cases?

Other concerns:
- The paper is lack of the report of GPU memory consumption. Since the sliding-window attention and AMF modules require storing multiple latent tensors, attention maps, and cached gradients, this may increase GPU memory usage (especially for long or high-resolution videos). However, the authors only provide runtime speedups without the profiling of memory consumption.
- The ablation on the selection of some important hyperparameter selection is also missing. For example, the window size and the gradient skip intervals could largely affect the balance / trade-off between visual quality and runtime. Could the authors also provide such ablation?

[1] "Vision Transformers Need Registers", ICLR 2024

**Questions:**

- How do you handle the irregular tiling that comes from the sliding window attention mechanism?
- Does the model use Flash Attention or other memory-efficient attention operations? If not, what is the runtime or memory overhead when switching to other attention functions?
- Could the authors provide the video outputs with longer durations or more complex motion (including both object motion and camera movement)?
- Since GPU memory consumption is also an important metric to evaluate the efficiency, could the authors also provide the report of memory usage?
- Could the authors include an ablation study for key hyperparameters, such as the window size, stride, temporal span, and gradient skip interval?

---

> ### Author Response · Authors · 2025-11-21
> **Feedback to Reviewer RmCa**
>
> Thanks for your time and thoughtful review. We appreciate your recognition of the satisfying experimental results and clear writing of the paper. We also add the experiment and details, and highlight using _**blue**_ color  in paper.  We provide our feedback as follows.
>
>
> > **Q1: Concern about Flash Attention and irregular attention**
>
>
> **A1**: Valuable question! in our experiment, During motion extraction, we don't use the flash attention. We only select the Q and K in the attention map and resize them into a 2D level to get the corresponding window. Then each Q is used to calculate similarity with K in corresoponding area to get the motion flow efficiently. We use zero padding to avoid the irregular issue. During the generation stage,  we use the flash attention for fast inference. We follow the same attention operation in DiTFlow (https://github.com/ditflow/ditflow/blob/main/guidance_utils/motion_flow_utils.py#L5 ) for fair comparion. These details will be added in the final version.
>
>
> > **Q2: longer sequences and larger motion**
>
> **A2**: Thanks, In the demo videos and main paper's figures, we are able to generate high-quality videos with consistent motion. However, We note that larger  and fast motion requir a larger window size and a temporal span for better performance.  Additionally, we perform the ablation in Tab. 3,4,5,7.  This limitation can be addressed by sparse sliding window and Dilated windows in the future. For more complex motion, our method performs strongly on both complex object motion and complex camera movement(As shown in Fig. 7 of Appendix). For object motion, it not only transfers the coarse trajectory of the object but also reproduces fine-grained, complex actions. For camera movement, it successfully transfers the full evolution and pacing of the motion. For the longer sequences, It depends on the  ability  of the basic fundational model. Now WAN-2.1 only supports 81 frames, which will be longer in the future. We will add the discussion  in the final version.
>
>
> **Tab.3(Appendix) Ablation study about temporal span.** Best results are in **bold**.
>
> | Temp. Span | Sub. Cons. | Back. Cons. | Aes. Qual. | Motion Smooth. |
> |---|---:|---:|---:|---:|
> | span-3 | 0.9592 | 0.9461 | *0.5690* | **0.9899** |
> | **span-5** | **0.9809** | **0.9684** | **0.5778** | *0.9891* |
> | span-7 | *0.9711* | *0.9608* | 0.5522 | 0.9858 |
>
>
> **Tab.4(Appendix) Ablation study about gradient skip interval.** Best results are in **bold**.
>
> | Skip Interval | Sub. Cons. | Back. Cons. | Aes. Qual. | Motion Smooth. |
> |---|---:|---:|---:|---:|
> | interval-1 | **0.9824** | **0.9695** | *0.5762* | *0.9853* |
> | **interval-2** | *0.9809* | *0.9684* | **0.5778** | **0.9891** |
> | interval-3 | 0.9651 | 0.9328 | 0.5521 | 0.9528 |
>
>
>
>
> **Tab.5(Appendix) Ablation study about window size.** Best results are in **bold**.
>
> | Wind. Size | Sub. Cons. | Back. Cons. | Aes. Qual. | Motion Smooth. |
> |---|---:|---:|---:|---:|
> | 17-size | 0.9625 | 0.9521 | **0.5680** | *0.9881* |
> | **21-size** | **0.9809** | **0.9684** | *0.5778* | **0.9891** |
> | 25-size | *0.9639* | *0.9550* | 0.5495 | 0.9860 |
>
>
> **Tab.7(Appendix) Ablation study about stride.** Best results are in **bold**.
>
> | Slid. Stride | Sub. Cons. | Back. Cons. | Aes. Qual. | Motion Smooth. |
> |---|---:|---:|---:|---:|
> | **stride-1** | **0.9809** | **0.9684** | **0.5778** | **0.9891** |
> | stride-3 | 0.9627 | *0.9630* | 0.5660 | *0.9875* |
> | stride-5 | *0.9629* | 0.9444 | *0.5699* | 0.9850 |
>
> > . **Q3: Report of GPU memory consumption**
>
> **A3**: Valuable suggestion!, We report  the GPU memory consumption in Sec.F of Appendix. We employ the Wan-2.1-14B as the same backbone for fair comparison. During the generation of 41‑frame videos, the peak GPU memory usage remained below 60 GB. This modest memory footprint demonstrates that our approach is GPU‑memory efficient, inference can be performed on a single 80‑GB GPU without memory pressure, facilitating broader accessibility and deployment in resource‑constrained environments.
>
> > **Q4: More ablation**
>
> **A4**: We ablate the key hyperparameters about window size, stride, temporal span, and gradient skip interval in Tabs. 3,4,5,7 of Appendix. We discover that a suitable gradient skip is beneficial for performance about motion fidelity and speed. Too large gradient skip result in the degradation of generated results.

---

### Official Review · Reviewer_TXvo · 2025-11-01

**Soundness:** 2
**Presentation:** 3
**Contribution:** 3
**Rating:** 6
**Confidence:** 3

**Summary:**

This paper addresses the inference speed bottleneck of Video Motion Transfer (VMT) that uses training-free Attention Motion Flow. Specifically, it employs,

(1) Sliding-window motion extraction, which assumes that motion correspondences between frames are local

(2) Step-skipping gradient optimization (reusing cached gradients), based on the observation that gradients across consecutive inner optimization steps are highly similar

The paper claims their proposed method achieves an average 3.43× speed-up without compromising visual fidelity or temporal consistency.

**Strengths:**

- The paper addresses a practical problem such as inference speed improvement in training-free motion transfer
- The proposed method also maintains (or even improves) the quality of the video
- Clearly identifies and analyzes existing problems such as motion and gradient redundancy through experiments

**Weaknesses:**

- The sliding-window strategy rests on the assumption that inter-frame motion is local and small. The paper lacks analysis or discussion of performance limitation when this assumption breaks (e.g., very fast and large motions, aggressive camera movements, occlusions).
- In Table 2, FastVMT uses WAN-2.1 as the base model, while other baselines may rely on different backbones. The paper states “fair backbone: WAN-2.1,” but it is unclear whether this means all baselines were re-implemented and re-evaluated on WAN-2.1, or merely that FastVMT used WAN-2.1. If the former, the results are very compelling. Otherwise, the quality advantage might partly stem from the newer backbone.
- The core acceleration idea of step-skipping is a fairly common design principle in other areas, so its novelty is somewhat limited.

**Questions:**

- In Table 2, are those baselines re-implemented by the authors using the WAN-2.1 backbone, or are these numbers quoted from the original papers (potentially with different backbones)?
- In Table 2, it looks like "Ours" is included under Tuning-Based Methods. Is that intentional?
- When is the window center re-estimated along the diffusion trajectory? Fixed at (t=0), or updated progressively during the first 20% of guided steps?

---

> ### Author Response · Authors · 2025-11-21
> **Feedback to Reviewer TXvo**
>
> Thank you for your comprehensive and detailed review of our paper and the recognition of our work's clarity and effectiveness. We also add the experiment and details, and highlight using _**blue**_ color in the paper.  We provide our feedback as follows.
>
> > **Q1: Limitation about sliding-window strategy**
>
> **A1**: Valuable question! Larger camera motion and fast motion require a larger window size and temporal span for better performance.  In our experiment, the stride, temporal span, and window size are hyperparameters.  Following the experiment setting in the main paper, we perform the ablation in Tab. 3, 4, 5, and Tab. 7. This limitation can be migrated by sparse sliding window and dilated windows in the future. We will add this part in the final version.
>
> **Tab.3(Appendix) Ablation study about temporal span.** Best results are in **bold**.
>
> | Temp. Span | Sub. Cons. | Back. Cons. | Aes. Qual. | Motion Smooth. |
> |---|---:|---:|---:|---:|
> | span-3 | 0.9592 | 0.9461 | *0.5690* | **0.9899** |
> | **span-5** | **0.9809** | **0.9684** | **0.5778** | *0.9891* |
> | span-7 | *0.9711* | *0.9608* | 0.5522 | 0.9858 |
>
>
> **Tab.4(Appendix) Ablation study about gradient skip interval.** Best results are in **bold**.
>
> | Skip Interval | Sub. Cons. | Back. Cons. | Aes. Qual. | Motion Smooth. |
> |---|---:|---:|---:|---:|
> | interval-1 | **0.9824** | **0.9695** | *0.5762* | *0.9853* |
> | **interval-2** | *0.9809* | *0.9684* | **0.5778** | **0.9891** |
> | interval-3 | 0.9651 | 0.9328 | 0.5521 | 0.9528 |
>
>
>
>
> **Tab.5(Appendix) Ablation study about window size.** Best results are in **bold**.
>
> | Wind. Size | Sub. Cons. | Back. Cons. | Aes. Qual. | Motion Smooth. |
> |---|---:|---:|---:|---:|
> | 17-size | 0.9625 | 0.9521 | **0.5680** | *0.9881* |
> | **21-size** | **0.9809** | **0.9684** | *0.5778* | **0.9891** |
> | 25-size | *0.9639* | *0.9550* | 0.5495 | 0.9860 |
>
>
> **Tab.7(Appendix) Ablation study about stride.** Best results are in **bold**.
>
> | Slid. Stride | Sub. Cons. | Back. Cons. | Aes. Qual. | Motion Smooth. |
> |---|---:|---:|---:|---:|
> | **stride-1** | **0.9809** | **0.9684** | **0.5778** | **0.9891** |
> | stride-3 | 0.9627 | *0.9630* | 0.5660 | *0.9875* |
> | stride-5 | *0.9629* | 0.9444 | *0.5699* | 0.9850 |
>
>
> > **Q2: Backbone question**
>
> **A2**: Thanks! In our experiment, we select the Wan-2.1-14B-T2V as the _**same backbone**_ for fair comparison and _**reimplement**_ previous work using Wan-2.1-14B-T2V.   The experiments are performed in three hyperparameters for fair comparison.  We report the average results finally.
>
>
>
> > **Q3: Core acceleration idea of Step-skipping**
>
> **A3**: Thanks. Step-skipping is a fairly common design in many areas. However, we find that motion transfer is a case of “stable gradient optimization” and _**aims to address gradient redundancy with the step-skipping strategy in gradient optimization**_, which is _**novel**_ and _**motivational**_ for the community.
>
> > **Q4: "Ours" is included under Tuning-Based Methods**
>
> **A4**: Thanks, our FastVMT is a training-free method, we update it to avoid misunderstanding.
>
> > **Q5: The timestep of the window center re-estimated**
>
> **A5**: In our experiments, the window center is updated progressively during the _**ALL**_ of guided steps.  This operation enables enhancing the computational efficiency and precision of AMF extraction.

---

### Official Review · Reviewer_sSov · 2025-11-01

**Soundness:** 3
**Presentation:** 3
**Contribution:** 2
**Rating:** 6
**Confidence:** 3

**Summary:**

The paper focuses on the inefficiency issue of motion transfer in the DiT architecture and proposes a method to improve it without sacrificing quality. Specifically, the authors point out that motion redundancy arises from the neglect of motion smoothness across frames, while gradient redundancy occurs due to ignoring the slow gradient changes along the diffusion trajectory. Accordingly, they propose a sliding-window strategy that operates on downsampled attention maps and a step-skipping gradient computation strategy, which together enhance the efficiency of motion transfer.

**Strengths:**

1. The paper is well written, clearly motivated, and easy to follow.
2. The proposed efficiency improvement strategies, involving the sliding-window and step-skipping gradient optimization, make sense to me, and the illustration of their rationale in the method section is intuitive.
3. The experimental results demonstrate that the proposed method maintains video quality while achieving notable speedup.

**Weaknesses:**

1. Evaluation dataset. The authors use 50 videos selected from the DAVIS dataset, which is rather small in scale and may not cover sufficient scene and motion diversity. I notice that benchmarks used in different motion transfer papers vary—perhaps the authors follow the test set of DiTFlow? However, how does the proposed method perform on other test sets used in related works? For example, please refer to Table 1 of DeT.

2. Implementation details. The authors mention that “for fair comparison, they adapt Wan-2.1 as the same backbone.” Is the 14B model or the 1.3B model used? Previous works adopt different backbones such as CogVideoX and Hunyuan. Are the authors reimplementing these methods using the Wan2.1 model? If so, there may be a risk that some methods cannot perform optimally, as they can be sensitive to hyperparameters. Overall, it would be helpful if the authors could provide more details about how each baseline is implemented.

3. It would be beneficial if the authors could include and analyze some typical failure cases of the proposed method.

**Questions:**

In the ablation study, adding the step-skipping strategy brings improvements in certain metrics such as aesthetics or text–frame similarity. However, I think the operation of reusing previous gradients at specific timesteps is essentially an approximation. Even if it does not cause a performance drop, it theoretically should not lead to improvement. Could the authors provide some explanation for this phenomenon?

---

> ### Author Response · Authors · 2025-11-21
> **Feedback to Reviewer sSov (1/2)**
>
> Thank you for your comprehensive review of our paper. We provide our feedback as follows. We also add the experiment and details, and highlight using _**blue**_ color  in paper.
>
> > **Q1: Evaluation dataset**.
>
> **A1**: In main paper, we follow the test set of DiTFlow to evaluate the FastVMT's performance. Additionally, _**we also provide the evaluation on the DeT's benchmark**_. The results are reported in Tab.6 (Appendix).  Our approach has better performance than previous works, further showing the superiority of FastVMT.
>
> **Tab.6-1(Appendix) Quantitative Metrics (MTBench)**
> | Method | Text Sim. ↑ | Motion Fid. ↑ | Temp. Cons. ↑ | Time (s) ↓ |
> |---|---:|---:|---:|---:|
> | **Training-Based Methods** |  |  |  |  |
> | MotionInversion | 0.2190 | *0.6945* | 0.9634 | 632.41 |
> | MotionDirector | *0.2351* | 0.6270 | 0.9599 | 806.64 |
> | DeT | 0.2317 | 0.5225 | 0.9609 | 2745.60 |
> | **Training-Free Methods** |  |  |  |  |
> | MOFT | 0.2238 | 0.5187 | 0.9375 | 595.81 |
> | MotionClone | 0.2161 | 0.5601 | *0.9775* | 397.05 |
> | SMM | 0.2112 | 0.5641 | 0.9468 | 809.70 |
> | DiTFlow | 0.2296 | 0.5126 | 0.9575 | *626.83* |
> | **Ours** | **0.2434** | **0.7182** | **0.9809** | **184.18** |
>
> **Tab.6-2(Appendix) VBench Metrics (MTBench)**
> | Method | Sub. Cons. ↑ | Back. Cons. ↑ | Aes. Qual. ↑ | Motion Smooth. ↑ |
> |---|---:|---:|---:|---:|
> | **Training-Based Methods** | |  |  |  |
> | MotionInversion | 0.9291 | *0.9587* | 0.4882 | 0.9658 |
> | MotionDirector | 0.9644 | 0.9435 | 0.3771 | 0.9650 |
> | DeT | 0.9540 | 0.9409 | 0.5063 | *0.9682* |
> | **Training-Free Methods** |  |  |  |  |
> | MOFT | 0.9471 | 0.9344 | 0.3672 | 0.9683 |
> | MotionClone | 0.9664 | 0.9550 | 0.5047 | 0.9660 |
> | SMM | *0.9720* | 0.9444 | 0.4554 | 0.9595 |
> | DiTFlow | 0.9402 | 0.9438 | *0.5156* | 0.9625 |
> | **Ours** | **0.9734** | **0.9690** | **0.5367** | **0.9781** |
>
> > **Q2: Implementation details.**
>
> **A2:**  Thanks for your suggestion! In our experiment, we select the Wan-2.1-14B-T2V as _**the same backbone**_ and _**reimplement**_ the previous work using the Wan2.1-14B for fair comparison.  The experiments are performed with three hyperparameters for fair comparison.  We report the average results finally. The details are as follows:
> - For MotionInversion, we follow the design of Motion Embeddings. The training stage is 1000 steps, and the learning rate is 1e-4. The denoising step is 50 and cfg is 7 in inference stage.
> - For MotionDirector, we use Wan-2.1-14B Lora tuning. The rank is 32. learning rate is 1e-4. We finetune them with 500 step both spatial and temporal training. During the inference, the denoising step is 50 and cfg is 7.5.
> - For DeT, we  train the model for 500 steps using the AdamW optimizer. The learning rate is 1e-5. We select the temporal kernel size as 5. The weight of dense point tracking loss is set to 256. The guidance scale is 7.5.
> - For DiTFlow, the optimization step is set to 20 steps. The denoising steps are employed for 50 steps, and we extract queries/keys from the 15th DiT block.
> - SMM: For SMM, we employ flow-matching-based inversion, rf-inversion to get the inversed noise. For space-time feature loss, we set the weight as 1.5. The scale range is from 0.005 to 0.002.
> - MotionClone: For given real videos, we apply a single denoising step of 500 for motion representation extraction. Guidance weight is 7 during inference. k = 1 is adopted for the mask. The denoising step is 50, while applying motion guidance in the first 15 steps
> - MOFT:  We employ flow-matching-based inversion, rf-inversion to get the inversed noise. The denoising step is 50 steps in our experiments.

---

> ### Author Response · Authors · 2025-11-22
> **Feedback to Reviewer sSov (2/2)**
>
> > **Q3: Failure cases**
>
> **A3**: Valuable suggestion!  As shown in previous works, current methods are still limited by the pre-trained video diffusion model's ability.  So it has difficulty transferring motion with prompts or motions that are out of distribution. For example, complex occlusions problems  (e.g., multi-person dance) still remain a difficult task for these models. Moreover, we highlight that motion transfer is inherently ambiguous if not associated to prompts. For example, transferring the motion of a human to a car may risk mapping motion features of other elements in the scene. (As shown in Fig.6 of the Appendix).
>
> > **Q4: Performance improvements of step-skipping**
>
> **A4**: Good question!  We analyze this issue of gradient redundancy during optimization, where _**over-optimization leads to performance degradation**_. We introduce skip-skipping to alleviate this problem and improves generation quality. We conduct detailed ablation studies on skip-skipping, as shown in the Tab. 4(Appendix). Moderate skipping helps boost the model performance, while excessive skipping results in degradation. We will include this analysis in the final version.
>
> **Tab.4(Appendix) Ablation study about gradient skip interval.** Best results are in **bold**.
> | Skip Interval | Sub. Cons. | Back. Cons. | Aes. Qual. | Motion Smooth. |
> |---|---:|---:|---:|---:|
> | interval-1 | **0.9824** | **0.9695** | *0.5762*| *0.9853* |
> | **interval-2** | *0.9809* | *0.9684* |**0.5778**| **0.9891**|
> | interval-3 | 0.9651 | 0.9328 | 0.5521 | 0.9528|

---

### Meta-Review · Area_Chair_EzAj · 2026-01-11

**Summary:**

Reviews were initially all positive, but did cite concerns about:

- Evaluation baselines and dataset: size of initial evaluation was too small.
- Fair comparison of backbones
- Lack of efficiency analysis.
- Visual quality
- Lack of more detailed ablation studies on window size, stride, span, etc.
- Limitations when generating longer sequences
- Limitations of sliding window on fast or large motions
- Novelty of step skipping

Most of these concerns were addressed---see next section.

**Reviewer Concerns:**

Among the listed concerns above, all but the following three seem to have been fully addressed:

- Limitations when generating longer sequences: The authors acknowledge that this is a limitation of current models and will improve in the future.
- Limitations of sliding window on fast or large motions: The authors acknowledge this limitation, but suggest that solutions to this may fall under the scope of future work.
- Novelty of step skipping: the authors argue that the particular way that this technique is applied is indeed novel.

**Reviewer Scores:**

Reviewer scores were initially all marginally positive, and one or two of them may have increased slightly after discussion.

---

### Decision · Program_Chairs · 2026-01-26

Accept (Poster)